

# Establishing isotopic turnover rates and trophic discrimination factors in tiger beetle (Coleoptera: Cicindelidae) larvae through a diet switch experiment

Lesa S. Giesbrecht[1,2], Aaron J. Bell[1,2], Timothy D. Jardine[3], Sean M. Prager[4] and Iain D. Phillips[2,5]

[1] Department of Biology, University of Saskatchewan, Saskatoon, Saskatchewan, Canada
[2] Troutreach Saskatchewan, Saskatchewan Wildlife Federation, Moose Jaw, Saskatchewan, Canada
[3] School of Environment and Sustainability, University of Saskatchewan, Saskatoon, Saskatchewan, Canada
[4] Department of Plant Sciences, University of Saskatchewan, Saskatoon, Saskatchewan, Canada
[5] Ecological and Habitat Assessment Services, Water Security Agency, Saskatoon, Saskatchewan, Canada

Corresponding author
Lesa S. Giesbrecht,
lesa.giesbrecht@usask.ca

## ABSTRACT

Stable isotope ratios give insight into food web interactions, but interpretation can be clouded by the timing of isotopic change associated with changes in diet and the difference in isotope ratios between consumers and their diets at equilibrium. The $^{15}N/^{14}N$, deemed $\delta^{15}N$, increases with each trophic transfer as $^{15}N$ becomes enriched, whereas the $^{13}C/^{12}C$ ratio, denoted as $\delta^{13}C$, remains relatively constant with each trophic transfer but can be influenced by lipid content. This study reports the trophic discrimination factors and isotopic half-lives in tiger beetles (Coleoptera: Cicindelidae). Wild-caught tiger beetle larvae were reared in a laboratory setting, subjected to a diet switch experiment, and sampled over time. Quadratic plateau models were used to characterize the change in $\delta^{15}N$, $\delta^{13}C$, and $\delta^{13}C_{corr}$ over time. Trophic discrimination factors were calculated by subtracting the mean prey $\delta^{15}N$, $\delta^{13}C$, and $\delta^{13}C_{corr}$ from that of the tiger beetle asymptotic $\delta^{15}N$, $\delta^{13}C$ and $\delta^{13}C_{corr}$ values, respectively. The tiger beetle trophic discrimination factor for $\delta^{15}N$ was 1.7 ± 0.2‰ with a half-life of 11.4 days. For $\delta^{13}C$, the trophic discrimination factor and half-life were –0.6 ± 0.2‰ and 3.9 days, respectively. After correcting for lipids ($\delta^{13}C_{corr}$), the trophic discrimination factor was –0.2 ± 0.2‰ with a half-life of 4.7 days. Isotopic turnover was fast with short half-lives, but factors that influence metabolic rates, such as ambient temperatures and life stage, should be considered when applying these estimates to wild tiger beetles. Despite this, the trophic discrimination factors and turnover rates calculated in this article are suitable estimates which can be applied to future studies.

## INTRODUCTION

Food webs are exceptionally complicated systems of energy flow between living organisms. By using ecological tracers such as stable isotopes, researchers can track this flow of energy and thus understand what trophic interactions are occurring in a given ecosystem (*Parnell et al., 2013*). This knowledge aids in conservation, especially in ecosystems with invasive species, as researchers are able to identify energy pathways for at-risk species and invasive species alike (*Muñoz, Currin & Whitfield, 2011*; *Jackson et al., 2012*; *Holthuijzen et al., 2023*). Heavier isotopes tend to become more enriched between trophic levels, a process termed trophic discrimination (*Bowes & Thorp, 2015*; *Veselý et al., 2024*). Stable isotopes are reported as a ratio of ratios using the following formula:

$$\delta^{15}N \text{ or } \delta^{13}C = [(R_{sample}/R_{standard}) - 1] \times 1000$$

where $R = {}^{15}N/{}^{14}N$ or ${}^{13}C/{}^{12}C$ (*Perkins et al., 2014*). The nitrogen-15 ratio ($\delta^{15}N$) increases with each upward step in the food chain and is used to determine an organism's trophic position (*Post, 2002*). The carbon-13 ratio ($\delta^{13}C$), however, usually remains relatively unchanged throughout the food chain compared to $\delta^{15}N$ and is commonly used to decipher the underlying energy pathways, such as the primary producer, of the particular food web that an organism is part of (*Post, 2002*; *Phillips et al., 2014*; *Zalewski et al., 2014*).

Most models assume $\delta^{15}N$ and $\delta^{13}C$ trophic discrimination factors (TDFs), which is the difference between the $\delta^{15}N$ or $\delta^{13}C$ of a consumer and its diet, to be 3.4‰ and 0.4‰, respectively (*Post, 2002*; *Bowes & Thorp, 2015*; *Quinby, Creighton & Flaherty, 2020*). Though isotope mixing models incorporate uncertainties associated with TDFs (*Phillips et al., 2014*), assuming a mean $\delta^{15}N$ TDF of 3.4‰ and a mean $\delta^{13}C$ TDF of 0.4‰ may result in incorrect conclusions. This is because TDFs often vary between taxa, and these assumed values were mostly derived from vertebrates and non-insect invertebrates (*Post, 2002*; *Veselý et al., 2024*). Additionally, $\delta^{13}C$ TDFs can be influenced by the physiological status of the organism because lipids synthesized *de novo* are depleted in ${}^{13}C$ (*DeNiro & Epstein, 1977*). This uncertainty is supported by several experiments with insects that have TDFs both higher and lower than the assumed $\delta^{15}N$ TDF of 3.4‰ and $\delta^{13}C$ TDF of 0.4‰ (*Traugott et al., 2007*; *Jardine et al., 2008*; *Perkins et al., 2014*; *Matos et al., 2018*). One caveat to this trend is phloem sap feeders, such as aphids, which tend to have negative TDFs (*Madeira et al., 2013*; *Perkins et al., 2014*; *Zhang et al., 2015*). By conducting stable isotope studies on organisms without first establishing species- or genus-specific TDFs, we risk inaccurate assessment of the basal food source of an organism's food web and the organism's trophic level, especially at higher trophic levels (*Perkins et al., 2014*; *Phillips et al., 2014*; *Quinby, Creighton & Flaherty, 2020*). For example, if an organism has a $\delta^{15}N$ TDF of 1.7‰, then its trophic position will be underestimated by half when using the commonly applied value of 3.4‰. To achieve more reliable trophic position estimates of a consumer, controlled feeding experiments are needed to understand potential deviations from typical TDFs (*Phillips et al., 2014*; *Quinby, Creighton & Flaherty, 2020*).

In addition to TDFs, isotopic half-lives can provide insight into an organism's metabolism and allow estimation of the duration over which the organism's diet is

represented (*Vander Zanden et al., 2015*). High tissue turnover rates indicate that the tissue or organism is highly metabolically active and/or that the organism is growing (*Sacramento, Manetta & Benedito, 2016*; *Li, Roth & Detwiler, 2018*). Half-lives range widely depending on the organism's size, temperature, and whether an organism is endo- or ectothermic (*Boecklen et al., 2011*; *Vander Zanden et al., 2015*). Tissue type is also a determinant of half-life, but whole-body half-lives are typically calculated in small organisms such as insects and small fish (*Vander Zanden et al., 2015*). *Vander Zanden et al. (2015)* developed a dataset of articles published between 1982–2014 in which they found four articles reporting whole organism half-lives of arthropod species. These four articles reported a range of 1.5 days (Hemiptera; *Jardine et al., 2008*) to 9.3 days (Collembola; *Larsen et al., 2011*) for $\delta^{15}$N, and 1.5 days (Diptera; *Overmyer, MacNeil & Fisk, 2008*) to 10 days (Collembola; *Larsen et al., 2011*) for $\delta^{13}$C. Unlike TDFs, there is no consistent difference between $\delta^{15}$N and $\delta^{13}$C half-lives reported in the literature (*Vander Zanden et al., 2015*).

Tiger beetles (Coleoptera: Cicindelidae) are a diverse, cosmopolitan beetle family that can be found in many habitat types (*Pearson, 1988*; *Duran & Gough, 2020*). As beetles, Cicindelids have a holometabolous life cycle with three larval instars (*Pearson, 1988*). Between instars and between third instar and pupation they molt, leaving behind a chitinous exoskeleton termed exuvia (*Tibbets, Wheeless & Del Rio, 2007*). Tiger beetles are carnivorous throughout their entire life cycle and are characterized by their fast terrestrial speeds and long, sickle-shaped mandibles (*Pearson, 1988*). They may become more prominent bioindicators in the coming years because they often occupy small, localized niches and are sensitive to changes in the environment (*Carroll & Pearson, 1998*; *Rodríguez, Pearson & Barrera, 1998*; *Svehla et al., 2023*). Because they occupy localized niches, several Canadian species and subspecies are at-risk, such as *Cicindela formosa gibsoni* WJ Brown, 1940, *Omus audouini* LJ Reiche, 1838, and *Cicindela marginipennis* PFMA Dejean, 1831 (*French et al., 2021*; *Klymko, 2021*; *McGregor, 2021*).

The purpose of this article is to provide insight into tiger beetle stable isotope TDFs and half-lives. This was done by subjecting wild-caught tiger beetle larvae to a diet switch experiment in a laboratory setting where they were sampled periodically. Stable isotope analysis was done on these samples to observe the isotopic turnover over time and to calculate tiger beetle TDFs. With this knowledge, researchers may apply the values determined here in future food web studies of wild tiger beetles and related insects. To our knowledge, this is the first study to conduct a diet switch on any tiger beetles for this purpose. Based on the available literature, we hypothesized that (1) the $\delta^{13}$C TDF will be a small but positive non-zero number, (2) the $\delta^{15}$N TDF will be below the commonly applied literature value of 3.4‰ but greater than zero, and (3) the $\delta^{15}$N and $\delta^{13}$C half-lives will be rapid in the fast-growing larval life stage and will be comparable to other invertebrates.

## MATERIALS AND METHODS

### Specimen collection

In this experiment, we chose to use three sympatric species as representatives: *Cicindela repanda* PFMA Dejean, 1825, *C. hirticollis* T Say, 1817, and *C. duodecimguttata* PFMA

Dejean, 1825. Henceforth, these three species will be referred to as tiger beetles. We chose to use multiple species rather than a single species for several practical reasons. First, larvae of these species are identified based on the number and position of setae on the pronotal disc (*Hamilton, 1925*) that are difficult to assess in the field. Secondly, the larvae were sympatric and occupied the same niche at the collection site, suggesting they likely had a similar diet in the wild (*Zalewski et al., 2014*). Finally, because the values calculated in this article are meant to apply to other cicindelids, we felt that any differences in turnover rates or TDFs between species would be small relative to differences among less closely related species (*Healy et al., 2018*).

First and second larval instars were collected from their native habitat along the South Saskatchewan River in Saskatoon, Saskatchewan, Canada (52°08′N 106°38′W). We collected larvae on 13–14 July 2023 using the "stab and grab" method outlined in *Brust, Hoback & Johnson (2010)*. This was done after sunset with flashlights so that the light reflected off the beetles' pronotal disc was evident making the larvae easy to locate. Once caught, larvae were stored in vials and brought back to the laboratory, where they were placed individually in 532 ml (18 oz) moulded polystyrene cups filled approximately two-thirds with moistened sand taken from their native habitat. The larvae were given 48 h to become habituated and to create new burrows. Based on a sub-sample ($n = 17$), 70.6% were *C. repanda*, 23.5% were *C. hirticollis*, and 5.9% were *C. duodecimguttata*.

## Laboratory experiment

The larvae were reared in a laboratory where their diet was switched from what they were eating in the wild to a controlled diet of dead *Trichoplusia ni* J Hübner, 1800–1803 caterpillars. The cups were kept in a growth chamber set to an 18:6 day (24 °C)-night (18 °C) cycle at 30% relative humidity. Every second day they were each fed a single dead *T. ni* caterpillar and watered with approximately 25–50 ml of deionized water using a wash bottle. The amount of water added was determined by how dry the substrate appeared visually. The *T. ni* themselves were reared on the McMorran diet so that their isotope ratios would remain constant throughout the study (*McMorran, 1965*).

Samples of the tiger beetle larvae, *T. ni*, and McMorran diet were collected throughout the experiment. We stored three to four larvae in a –80 °C freezer on days 0, 4, 6, 8, 14, 23, and 36 after field collection for a total of 27 larvae. Additionally, tiger beetle pupae ($n = 4$), larval exuviae of third instars ($n = 8$), and adults (upon emergence; $n = 10$) were also sampled and frozen. Sample sizes were limited due to the small population size at our collection site (Table 1). Samples were dried in an oven set to 60 °C for 24–48 h or lyophilized for 48 h, depending on equipment availability. Once dried, we crushed and subsampled 1.1 mg (±0.1 mg SD) for animal tissue and 3.3 mg (±0.8 mg SD) for the McMorran diet. Two samples per adult were taken, analyzed, and averaged for statistical analysis to ensure accurate values while avoiding pseudo-replication (*Quinby, Creighton & Flaherty, 2020*).

Subsamples were sent to the National Hydrology Research Centre (Saskatoon, Saskatchewan, Canada), where they were combusted in a Carlo Erba NA1500 Elemental Analyzer (ThermoFisher Scientific, Waltham, MA, USA). The vaporized $N_2$ and $CO_2$ were

**Table 1 Mean (±standard deviation (SD)) isotope ratios of *Trichoplusia ni* and the tiger beetles.**

|  | $\delta^{15}$N (‰) | $\delta^{13}$C (‰) | $\delta^{13}$C$_{corr}$ (‰) |
|---|---|---|---|
| *T. ni* larvae (*n* = 12) | 5.3 ± 0.6 | −23.7 ± 0.4 | −22.9 ± 0.5 |
| Wild-caught tiger beetle larvae (*n* = 4) | 9.0 ± 0.6 | −26.7 ± 1.5 | −26.0 ± 1.4 |
| Post-diet switch tiger beetle larvae (*n* = 7) | 7.0 ± 0.3 | −24.2 ± 0.3 | −23.1 ± 0.3 |
| Tiger beetle pupae (*n* = 4) | 8.0 ± 1.5 | −24.6 ± 0.9 | −23.4 ± 1.0 |
| Tiger beetle exuviae (*n* = 8) | 8.1 ± 0.6 | −25.1 ± 0.8 | −24.2 ± 1.0 |
| Tiger beetle adults (*n* = 10) | 7.4 ± 0.2 | −24.4 ± 0.3 | −23.6 ± 0.2 |

separated using gas chromatography and fed into a Delta V Isotope Ratio Mass Spectrometer (ThermoFisher Scientific, Waltham, MA, USA). The reference material used was Vienna Pee Dee Belemnite for $\delta^{13}$C and atmospheric nitrogen for $\delta^{15}$N. The isotope ratio mass spectrometer was calibrated using the bowhead whale baleen III keratin ($\delta^{13}$C = −20.2‰, $\delta^{15}$N = 14.3‰) and Pugel ($\delta^{13}$C = −13.6‰, $\delta^{15}$N = 5.1‰) internal laboratory calibration standards. Measurement precision was estimated to be ±0.1‰ for both isotopes.

To measure the health of reared adults and ensure the beetles grew and developed normally, we took elytron lengths and widths of seven *C. repanda*, three *C. hirticollis*, and one *C. duodecimguttata* adults that emerged at the end of the experiment and compared them to measurements of wild-caught adults provided by the Water Security Agency (Saskatoon, Saskatchewan, Canada). This was done by taking photos of the tiger beetle abdomens using an Axiocam 105 colour camera (Carl Zeiss Microscopy GmbH) and ZEN blue edition software version 2.3 (*Carl Zeiss Microscopy GmbH, 2016*) attached to a Zeiss Stereo Discovery V8 microscope (Carl Zeiss Microscopy GmbH). The measurements were then recorded using ImageJ version 1.54g (*Schneider, Rasband & Eliceiri, 2012*). The mean elytron lengths and widths of lab-raised *versus* wild-caught *C. repanda* and *C. hirticollis* adults were compared using Welch's t-tests. *C. duodecimguttata* was not tested because we only had a single lab-raised specimen.

## Data analysis

High lipid content (indicated by high C/N ratios) may drive $\delta^{13}$C values in a negative direction (*Jardine et al., 2008*; *Logan et al., 2008*). C/N ratios were, on average, greater than 4, above the value deemed to falsely skew $\delta^{13}$C by *Post et al. (2007)*. C/N ratios also increased over the course of the experiment from a mean of 4.1 on day 0 to 6.2 on day 37. Therefore, to correct for varying lipid levels between insect life stages, we employed the following invertebrate-specific formula provided by *Logan et al. (2008)*:

$$\delta^{13}C_{corr} = \delta^{13}C - (2.056 - 1.907 \times ln(C \div N)).$$

All data were analyzed in R version 4.3.1 using the package rcompanion (*Mangiafico, 2024*) for the quadratic plateau models (*R Core Team, 2024*). We used the quadratic plateau models to fit the isotope data to visualize the rate of change in larvae over time (*Mangiafico, 2016*). This model is similar to a linear plateau model, except the linear

portion is replaced with a quadratic term. Therefore, the quadratic portion and plateau portion, respectively, were calculated by:

$$y = -0.5 \times b \div clx$$
$$y = a + b \times clx - 0.5 \times b \times clx$$

where $a$ = the best-fit intercept, $b$ = slope, and $clx$ = critical $x$ value, or in other words, the day that the plateau is reached. The models were tested using the Cox and Snell pseudo-$R^2$ test (*Mangiafico, 2016*).

The half-lives were calculated using the following formula:

$$\text{Half-life} = ln(0.5)/\lambda$$

where $\lambda$ = turnover rate of $\delta^{15}N$, $\delta^{13}C$, or $\delta^{13}C_{corr}$ (*Hobson & Clark, 1992*). The turnover rate ($\lambda$) was calculated using the formula:

$$Y = Y_c + c \times e^{(-\lambda \times clx)}$$

where $Y$ is the stable isotope ratio over time, $Y_c$ is the asymptotic value, and $c$ is the difference between the initial isotope ratio and the ratio at equilibrium (*i.e.*, the asymptote value; *Sacramento, Manetta & Benedito, 2016*). The TDFs were calculated by subtracting the mean stable isotope ratios of the diet (*T. ni*) from the ratios of the at-equilibrium tiger beetle larvae (*i.e.*, the asymptotic plateau value; *Jardine et al., 2008*).

To measure any significant differences in mean $\delta^{15}N$, $\delta^{13}C$, and $\delta^{13}C_{corr}$ that may have occurred during pupation or emergence, we conducted an analysis of variance (ANOVA) using larvae (at equilibrium), pupae, exuviae, and adults. *Post hoc* Tukey honestly significant difference (HSD) tests were used to check which means were significantly different. We also tested whether post-diet switch larvae were significantly different from their diet in $\delta^{15}N$, $\delta^{13}C$, and $\delta^{13}C_{corr}$ (*i.e.*, whether TDF = 0) using one-sample t-tests.

## RESULTS

The tiger beetle larvae were reared successfully in the laboratory. The C/N ratio of the sampled larvae increased over time, with a 43% increase between day 0 and day 36 (Fig. 1A). The increase was linear, with an $R^2$ value of 0.66 ($P < 0.001$). Of the individuals that were not sampled as larvae ($n = 29$), 93% became pupae, and of those pupae that were not sampled ($n = 23$), 96% emerged as adults. The first adults emerged 70 days after we initially collected them from the wild, and the last beetle emerged on day 92. All three species of lab-raised adults were smaller on average compared to wild-caught adults, although these size differences were not significant (Table 2). The mean elytron length and width were 3.7% ($P = 0.080$) and 2.2% ($P = 0.421$) shorter, respectively, for lab-raised *C. repanda*. For lab-raised *C. hirticollis*, the mean length was 5.8% ($P = 0.252$) shorter, and the mean width was 0.2% shorter ($P = 0.973$). Finally, the single lab-raised *C. duodecimguttata* elytron length was 8.1% shorter in length and 5.4% shorter in width.

The nitrogen and lipid-corrected carbon isotopes, but not uncorrected carbon isotopes, varied between tiger beetle larvae and exuviae (Table 1). We found an effect of life stage for $\delta^{15}N$ ($F = 4.4$, $P = 0.012$), although this difference was only significant between larvae and

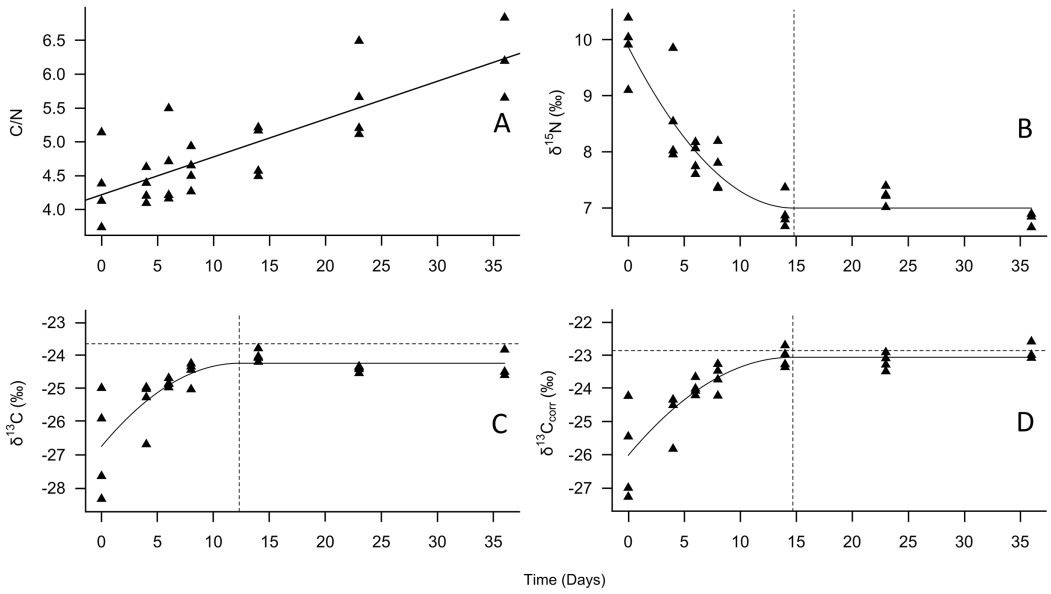

**Figure 1 Change in tiger beetle larval (A) C/N, (B) $\delta^{15}$N, (C) $\delta^{13}$C, and (D) $\delta^{13}$C$_{corr}$ ratios over the course of a diet switch experiment.** The best-fit line for (A) is a linear regression model to visualize the increase in carbon relative to nitrogen over time as the larvae grew. The best-fit line for (B), (C), and (D) were plotted using quadratic plateau models to visualize the change in larval tiger beetle (triangles, $n = 27$) stable isotope ratios over the course of a diet switch experiment. The $\delta^{13}$C$_{corr}$ ratios are the carbon ratios after being mathematically corrected for lipids. The vertical dashed lines in (B), (C), and (D) represent when the larvae reached isotopic equilibrium with their food source, *Trichoplusia ni* caterpillars, whose mean isotopic ratios are represented by the horizontal dotted lines in (C) and (D) to emphasize how little $\delta^{13}$C and $\delta^{13}$C$_{corr}$ changed between trophic levels. Trophic discrimination factors are the mean isotope ratios of *T. ni* subtracted from the plateau values.

**Table 2 Mean (±SD) elytra length and width of lab-raised and wild-caught adult tiger beetles.**

|  | Mean length (µm) | Mean width (µm) |
|---|---|---|
| *C. repanda* | | |
| Lab-raised ($n = 6$) | 346.3 ± 5.6 | 116.5 ± 4.2 |
| Wild-caught ($n = 10$) | 359.4 ± 20.1 | 119.1 ± 8.3 |
| *C. hirticollis* | | |
| Lab-raised ($n = 3$) | 376.3 ± 24.3 | 120.8 ± 11.4 |
| Wild-caught ($n = 10$) | 398.7 ± 26.7 | 121.1 ± 10.0 |
| *C. duodecimguttata* | | |
| Lab-raised ($n = 1$) | 348.5 | 113.5 |
| Wild-caught ($n = 10$) | 377.9 ± 19.1 | 119.8 ± 7.6 |

exuviae ($P = 0.010$, Fig. 2A) where the larvae were depleted in [15]N compared to exuviae. Similarly, there was also an effect of life stage for $\delta^{13}$C$_{corr}$ ($F = 4.1$, $P = 0.017$) with the difference also being between larvae and exuviae ($P = 0.015$, Fig. 2B) but the larvae were enriched in [13]C compared to the exuviae. We found no effect of life stage for $\delta^{13}$C ($F = 2.4$, $P = 0.089$).

According to the models, the asymptotic values were: for $\delta^{15}$N, 7.0‰ (±0.3‰ SD), which was achieved on day 14.8, for $\delta^{13}$C –24.2‰ (±0.3‰ SD), achieved on day 12.3, and

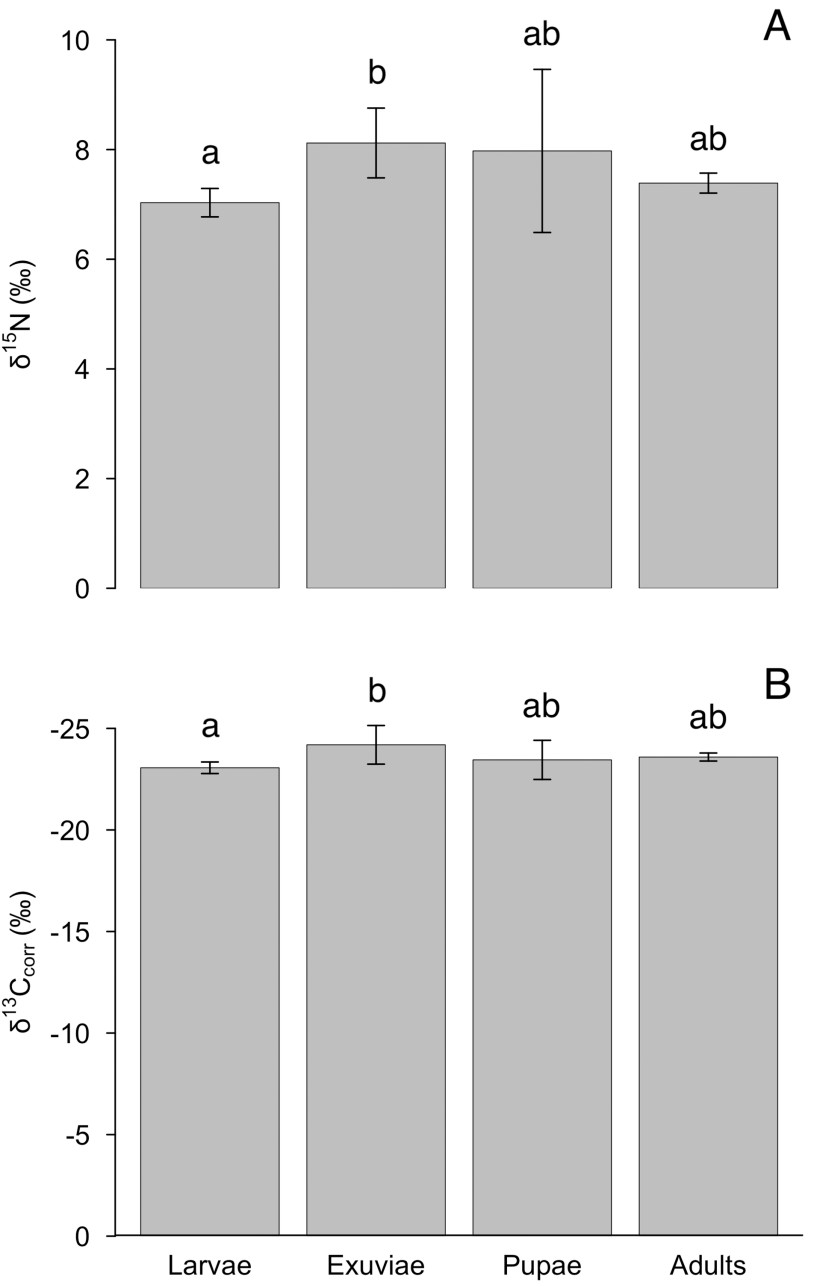

**Figure 2  Differences in mean (A) $\delta^{15}$N and (B) $\delta^{13}$C$_{corr}$ between tiger beetle life stages and exuviae at equilibrium.** The samples were taken from specimens that had reached isotopic equilibrium with their food source, *Trichoplusia ni* caterpillars, meaning that differences in isotopic ratios were not due to differences in diet. *Post hoc* Tukey HSD tests revealed that the exuviae (*n* = 8) were enriched in $\delta^{15}$N (*P* = 0.010) but depleted in $\delta^{13}$C$_{corr}$ (*P* = 0.015) relative to the larvae (*n* = 7). However, the larvae and exuviae were not significantly different from pupae (*n* = 4) and adults (*n* = 10). Error bars represent standard deviation, and lower-case letters indicate significance between means.

for $\delta^{13}C_{corr}$, −23.1‰ (±0.3‰ SD), achieved on day 14.6 (Fig. 1, Table 1). The Cox and Snell pseudo-$R^2$ values were 0.85, 0.66, and 0.66 for the $\delta^{15}N$, $\delta^{13}C$, and $\delta^{13}C_{corr}$ models, respectively. When the mean *T. ni* ratios (Table 1) were subtracted from the asymptotic values, the $\delta^{15}N$ TDF was 1.7 ± 0.2‰, the $\delta^{13}C$ TDF was −0.6 ± 0.2‰, and the TDF for $\delta^{13}C_{corr}$ was −0.2 ± 0.2‰. The TDF was significantly different from zero for $\delta^{15}N$ ($F = 48.2$, $P < 0.001$) and for $\delta^{13}C$ ($F = 18.2$, $P < 0.001$), but not $\delta^{13}C_{corr}$ ($F = 0.9$, $P = 0.352$). The half-life for $\delta^{15}N$ was 11.4 days, for $\delta^{13}C$, it was 3.9 days, and finally, the half-life for $\delta^{13}C_{corr}$ was 4.7 days.

## DISCUSSION

We found support for our predictions that the $\delta^{15}N$ TDF in tiger beetle larvae is lower than the literature-assumed value of 3.4‰ and that the turnover of $^{15}N$ and $^{13}C$ in tiger beetles occurs quickly (*i.e.*, days). Similarly, our $\delta^{13}C$ and $\delta^{13}C_{corr}$ half-lives of 3.9 and 4.7 days, respectively, fall in the center of the literature range of 1.5–10 days for whole-bodied arthropods, whereas our $\delta^{15}N$ half-life lies just above ranges reported in the literature (*Vander Zanden et al., 2015*). Our prediction of a small positive number for $\delta^{13}C$ was not supported as we calculated TDFs of −0.6 ± 0.2‰ for $\delta^{13}C$ and −0.2 ± 0.2‰ for $\delta^{13}C_{corr}$, with the latter not significantly differing from zero.

The $\delta^{15}N$ TDF calculated in our study yielded a smaller value than what has been used for isotope mixing models in the literature (*Post, 2002*; *Phillips et al., 2014*). Our value of 1.7 ± 0.2‰ is in line with reported TDFs for Elateridae ($\delta^{15}N$ TDF 1.1 ± 0.3‰ to 1.6 ± 0.2‰) but not for other beetles such as Coccinellidae (2.9‰), illustrating the importance of establishing taxon-specific TDFs (*Ostrom, Colunga-Garcia & Gage, 1997*; *Traugott et al., 2007*; *Quinby, Creighton & Flaherty, 2020*). Although the TDF for $\delta^{13}C$ (−0.6 ± 0.2‰) was slightly negative, it was consistent with values reported by *Jardine et al. (2008)* and fell within the typical range of ±3‰ reported by *Post (2002)* After lipid correction, there was no difference between the beetles and their diets, suggesting that the negative TDF for uncorrected $\delta^{13}C$ was driven by lipid content, as indicated by C/N ratios in beetles as high as 6.5 by the time they reached equilibrium with the new diet. The increase in fat stores may reflect the energy needed to overwinter, pupate, and avoid internal freezing (*Jardine et al., 2008*; *Enriquez et al., 2022*).

Future studies on wild tiger beetle food webs may use our reported $\delta^{15}N$ value of 1.7 ± 0.2‰ to determine a more accurate diet composition and trophic position than if one were to use the traditional assumption of 3.4‰ (*Bowes & Thorp, 2015*; *Morente & Ruano, 2022*). The assumed TDF of 0.4‰ for $\delta^{13}C$ should be replaced with a value of zero if lipid correction is applied, meaning that tiger beetles should closely match those of their prey's underlying sources of primary production.

Tissue turnover rates are variable based on metabolism and growth rate, and thus, would change at different temperatures, and in adults as they cease to grow post-eclosion (*Fry & Arnold, 1982*; *Sacramento, Manetta & Benedito, 2016*; *Li, Roth & Detwiler, 2018*). Wild tiger beetles likely consume food less frequently than the larvae we reared in the laboratory which would translate to a slower growth rate in wild specimens and therefore slower turnover rates (*Pearson & Knisley, 1985*). Regarding temperature, lower or higher

temperatures than our experimental 18–24 °C range would lead to slower or faster turnover rates, respectively, due to how temperature influences metabolism (*Acar et al., 2001*). Therefore, if our half-lives of 11.4 days for $\delta^{15}N$, 3.9 days for $\delta^{13}C$, and 4.7 days for $\delta^{13}C_{corr}$ are to be used in future research, temperature and life stage must be considered. Nonetheless, our reported half-lives for $\delta^{13}C$ and $\delta^{13}C_{corr}$ are well within the arthropod half-life range of 1.5–10 days found in *Vander Zanden et al. (2015)*, and our $\delta^{15}N$ half-life is only slightly above their longest reported half-life of 9.3 days, but far below the half-lives of some vertebrate tissue. This is consistent with our hypothesis that tiger beetle larvae have rapid stable isotope half-lives because larvae are small and grow rapidly. These fast turnover rates may allow researchers to track recent dietary shifts in wild tiger beetle populations (*Vander Zanden et al., 2015*).

Past studies have found that adult holometabolous insects, apart from beetles (*Tibbets, Wheeless & Del Rio, 2007*), are enriched in $^{15}N$ and depleted or unchanged in $^{13}C$ compared to larvae and pupae (*Barriga et al., 2013*; *Matos et al., 2018*). Our findings that $\delta^{15}N$, $\delta^{13}C$, and $\delta^{13}C_{corr}$ did not vary between tiger beetle adults, larvae, and pupae are consistent with *Tibbets, Wheeless & Del Rio (2007)* and support the application of our TDFs ($\delta^{15}N$: 1.7‰ and $\delta^{13}C$: 0‰) to both larval and adult tiger beetles. Hypothetically, this may be because the beetle species tested in *Tibbets, Wheeless & Del Rio (2007)* and our study do not excrete meconium, a metabolic waste product, prior to pupation, which tends to be depleted in $^{15}N$, resulting in adults being enriched. Alternatively, beetle meconium may simply not be depleted in $^{15}N$, which was not tested here or in *Tibbets, Wheeless & Del Rio (2007)*. Or, more simply, it may have been due to our small sample sizes. Our results for the exuviae, which were significantly enriched in $^{15}N$ compared to larvae, are consistent with *Tibbets, Wheeless & Del Rio (2007)* but not with *Traugott et al. (2007)*, who found no significant differences between exuviae and larval segments. *Traugott et al. (2007)* also found no differences between larval segment $\delta^{13}C$ and exuviae $\delta^{13}C$, which is consistent with our findings. Conversely, after correcting for lipids, $^{13}C$ was significantly depleted in our exuviae compared to larvae. Future research should test stable isotope ratios before, during, and after pupation on other beetle species, including sampling exuviae and meconium, to fill in these knowledge gaps.

The methods used to collect and rear tiger beetles were effective for species that live by the edge of a large river. The larvae were able to pupate without having to overwinter as third instars as typically occurs under natural conditions (*Hamilton, 1925*). Although on average the lab-raised adults were smaller than wild-caught adults the difference was not statistically significant. This was likely due to our small sample size of lab-raised tiger beetles. The high frequency of successful emergence demonstrates the efficacy of the method used in our experiment.

Tiger beetles have the potential to become a useful bioindicator of ecosystem health because they are globally distributed, yet at the species level, they occupy habitats that are sensitive to change (*Rodríguez, Pearson & Barrera, 1998*; *Cassola & Pearson, 2000*). Currently, however, only a handful of studies have used them in this capacity (*Carroll & Pearson, 1998*; *Svehla et al., 2023*). The values reported here may be used in future studies

examining tiger beetle feeding ecology and food web interactions as indicators of habitat and community health. To make accurate extrapolations about tiger beetle food web ecology, potential prey and primary producers should be analyzed alongside the tiger beetles (*Phillips et al., 2014*). This is so that energy pathways can be matched to the $\delta^{13}$C of tiger beetles (*Zhang et al., 2015*). This same principle applies to identifying tiger beetle prey and trophic position using $\delta^{15}$N (*Phillips et al., 2014*).

## CONCLUSIONS

The results from this experiment supported two of our three predictions. The $\delta^{15}$N TDF of 1.7‰ is indeed lower than the 3.4‰ assumed in most literature models. Conversely, the prediction that we made about the $\delta^{13}$C TDF being a small, positive and non-zero value was not met. However, our values still fell within the wide range of literature values for $\delta^{13}$C TDFs, and correcting for lipids brought values closer to zero. Finally, the $^{15}$N and $^{13}$C half-lives in our study are comparable to other invertebrate studies. The values calculated in our study may be used to better understand the feeding ecology of other tiger beetle species and their food web dynamics.

## ACKNOWLEDGEMENTS

We thank the members of the Prager Lab and Troutreach for helping to collect the larvae. We also specifically thank Caleb Bryan and Jeremy Irvine for assisting with statistics, as well as Alicia Caplan, Evan Quick, and Ningxing Zhou for feeding and watering the larvae. Thank you to the Frost Lab for reviewing the final draft, including John Acorn, who also helped us better understand tiger beetle lifecycles, which improved our experimental setup. Finally, we would like to thank the two reviewers for their generous feedback.

### Funding

This project was funded by EcoCanada's 2023 Co-Op Internship, Natural Science and Engineering Research Council of Canada's Research Discovery Grant Program (No. RGPIN-2018-04954), the Royal Bank of Canada Tech for Nature Grant (No. ENV20228170), and Service Canada through the Canada Summer Jobs Project (No. 019884097). The funders had no role in study design, data collection and analysis, decision to publish, or preparation of the manuscript.

### Grant Disclosures

The following grant information was disclosed by the authors:
EcoCanada's 2023 Co-Op Internship.
Natural Science and Engineering Research Council of Canada's Research Discovery Grant Program: RGPIN-2018-04954.
Royal Bank of Canada Tech for Nature: ENV20228170.
Service Canada through the Canada Summer Jobs Project: 019884097.

## Competing Interests

The authors declare that they have no competing interests.

## Author Contributions

- Lesa S. Giesbrecht conceived and designed the experiments, performed the experiments, analyzed the data, prepared figures and/or tables, authored or reviewed drafts of the article, and approved the final draft.
- Aaron J. Bell conceived and designed the experiments, analyzed the data, authored or reviewed drafts of the article, and approved the final draft.
- Timothy D. Jardine conceived and designed the experiments, analyzed the data, authored or reviewed drafts of the article, and approved the final draft.
- Sean M. Prager conceived and designed the experiments, authored or reviewed drafts of the article, and approved the final draft.
- Iain D. Phillips conceived and designed the experiments, analyzed the data, authored or reviewed drafts of the article, and approved the final draft.

## Data Availability

The data is available on Dryad: Giesbrecht, Lesa S.; Bell, Aaron J.; Jardine, Timothy D. et al. (2025). Supporting data for tiger beetle stable isotope analysis [Dataset]. Dryad. https://doi.org/10.5061/dryad.jh9w0vtmv.

The code is available at Zenodo: Giesbrecht, L. S., Bell, A. J., Jardine, T. D., Prager, S. M., & Phillips, I. D. (2025). Supporting data for tiger beetle stable isotope analysis. Zenodo. https://doi.org/10.5281/zenodo.14606358.

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
