# Peer review of "Establishing isotopic turnover rates and trophic discrimination factors in tiger beetle (Coleoptera: Cicindelidae) larvae through a diet switch experiment"

_PeerJ, doi:10.7717/peerj.19279_

## Round 0.1 · original submission · Minor Revisions

Dear colleagues, your article is very interesting and very well written. I think that correcting minor flaws will not take you much time and will allow you to publish this article as quickly as possible.

·

Basic reporting

- Methods: Please elaborate/clarify what is meant by ‘effect’ in lines 227, 230, 232.
- Line 152 and 208: This is the first time ‘exuviae’ appears. I think it would be helpful to describe a little bit more about tiger beetle life stages and lifecycle in the introduction, Paragraph lines 97-16, to provide sufficient field background/context. I had to look up exuviae to understand what you were talking about.
- Line 319: isotopes not formatted correctly; nitrogen-15 as opposed to N15

Experimental design

- It would be helpful either in the methods or results section to describe the total number of individuals used for the experiment. I do see it comes up in Table 2 but having an idea of the sample size as you’re reading in-text would help to better understand the experimental design.
-Line 180: You state that high lipid content can influence deltaC values and that this is indicated by high C/N ratios. Looking at Fig 1A and reviewing the results section, I think it would be valuable to provide more of an explanation on why you determined your C/N ratios were high and required lipid correction in either the methods or results (not just line 261 in the discussion). Some researchers argue that removing lipids can influence interpretation of consumer foraging habits and the literature is pretty split between those that do and do not apply lipid corrections. If you are going to use Logan’s lipid correction equation, I would just want to see a more solid explanation of why that was necessary and used in this study, particularly if you recommend replacing the TDF for C with 0 when lipid-corrected.

Validity of the findings

-Line 270: You state tissue turnover rates vary with metabolism and temperature and make note to consider that in future studies. What I would like to see addressed in this section, maybe in the paragraph from 298-301 is how the growth rates that differed between your lab-raised beetles and wild beetles may have influence empirically determined SIR and tissue turnover rates vs what would be seen in the wild. Because you present that information in table 1, a discussion on how wild vs lab raised specimens SIR may differ is warranted.

Reviewer 2 ·

Basic reporting

.

Experimental design

.

Validity of the findings

.

Additional comments

The article is very interesting and useful. The literature review is very comprehensive, the methods are described in detail and correspond to the objectives. The results are presented clearly, the conclusions seem adequate and beyond doubt.
A few minor comments:
53 repetition of the words "of a ratio"
72 floem feeders – aphids feed on plant sap, not phloem
139 Perhaps the sample is too small. If the samples are increased, the difference between wild and laboratory-grown beetles may become reliable (221), and the differences in the isotope ratios between larvae and adults may also be reliable (Figure 2).
Fig. 1. What do the vertical dotted lines mean?
Overall, the article looks ready for publication after eliminating minor comments.

---

## Round 0.2 · accepted · Accept

Dear authors, I am pleased to inform you that your article has been accepted for publication in our journal. I ask you to continue researching this very interesting topic.

·

Basic reporting

pass

Experimental design

pass

Validity of the findings

pass

Additional comments

Fantastic revisions. Accepted.

Reviewer 2 ·

Basic reporting

..

Experimental design

.

Validity of the findings

.

Additional comments

I have read the new version of the article and the authors' responses to the comments. The new version seems more correct to me and does not require further changes. In my opinion, the article is now ready for publication.